# Comparison of Simple Stylet versus Lighted Stylet for Intubating the Trachea with a Direct Laryngoscope: A Randomized Clinical Trial

**DOI:** 10.3390/jcm8020140

**Published:** 2019-01-25

**Authors:** Seongjoo Park, Jeongpyo Hong, Jin-Woo Park, Sung-Hee Han, Jin-Hee Kim

**Affiliations:** 1Department of Anesthesiology and Pain Medicine, Seoul National University Bundang Hospital, Seongnam-si, Gyeonggi-do 13620, Korea; struka@snubh.org (S.P.); 1gh404on3@naver.com (J.H.); jinul8282@snubh.org (J.-W.P.); anesthesiology@snubh.org (S.-H.H.); 2College of Medicine, Seoul National University, Jongno-gu, Seoul 03080, Korea

**Keywords:** airway management, direct laryngoscopy, lighted stylet

## Abstract

This study investigated the effectiveness of a lighted stylet during tracheal intubation with direct laryngoscopy. The study randomly assigned 284 patients undergoing general anesthesia to either the simple stylet (Group S) or lighted stylet (Group L) groups. In both groups, stylet-assisted intubation was performed with direct laryngoscopy. In group S, a simple stylet was used and removed when the tip of the endotracheal tube was thought to have passed the larynx. In Group L, a lighted stylet was used and removed after confirming transillumination of the suprasternal notch. The success rate at the first attempt, total intubation time, incidence of mucosal bleeding, and severity of postoperative sore throat were compared. Compared to a simple stylet, the lighted stylet significantly increased the success rate of tracheal intubation at the first attempt (128 (90%) vs. 140 (99%), *p* = 0.003, Groups S and L, respectively). The incidence of mucosal bleeding was significantly higher in Group S (35 (25%) vs. 19 (13%), *p* = 0.011, Groups S and L, respectively). The total intubation time and degree of postoperative sore throat were not significantly different between the two groups. A lighted stylet increased the success rate of tracheal intubation during stylet-assisted tracheal intubation with direct laryngoscopy.

## 1. Introduction

Despite more than 100 years of direct laryngoscopy, prompt and accurate placement of the endotracheal tube remains a considerable challenge in some patients-even in experienced hands. Failure or delay of endotracheal intubation may lead to a detrimental complication or death. To solve this problem, intubation devices with different blade designs, such as the McCoy leverage blade, or allowing an indirect view of the glottis, such as the fiberoptic bronchoscope and the video laryngoscope, have been designed. However, direct laryngoscopy remains the standard method for tracheal intubation in clinical practice.

The stylet modifies the shape of the endotracheal tube and facilitates the entrance of the tip into the larynx in both easy and difficult cases [1]. However, it is limited by the blind technique and uncertainty in the location of the tube tip, especially in difficult intubation, and a risk of failure always exists. Transillumination of suprasternal soft tissue makes it possible to verify the position of the lighted stylet and guide the endotracheal tube into the trachea [2]. However, no information is available on the effectiveness or feasibility of a lighted stylet while inserting a direct laryngoscope.

We hypothesized that a lighted stylet would facilitate the detection of the endotracheal tube tip in the trachea during intubation with a direct laryngoscope. Therefore, we compared a lighted stylet with a simple stylet in terms of the initial intubation success rate and the frequency of airway complications.

## 2. Experimental Section

After acquiring approval from the Institutional Review Board of the Seoul National University Bundang Hospital (B-1501-284-004; approval date: 2015/02/05) and registering the study with the Clinical Research Information Service (CriS; http://cris.nih.go.kr; registration number: KCT0001759), written informed consent to participate in this randomized observational study was obtained from each patient. This study was conducted from March 2015 to March 2016. The study enrolled 284 patients over 18 years old, all with an American Society of Anesthesiologists (ASA) physical class I–II who were scheduled for elective surgery under general anesthesia (neuro-intervention, *n* = 124; hand, *n* = 57; breast, *n* = 33; abdominal, *n* = 27; plastic, *n* = 26; thyroid, *n* = 7; cardio-intervention, *n* = 4; foot, *n* = 2; knee, *n* = 2; and eye, *n* = 2). Patients with an ASA class of III or above, dentures, a history of cervical spine surgery, cervical herniation of the intervertebral disc, psychological disease with or without psychoactive drug use, or craniofacial anomalies; who were at risk of difficult mask ventilation or pulmonary aspiration; and/or who needed fiberoptic intubation or a double lumen endotracheal tube, were excluded from this study. After enrollment, airway parameters were checked in all patients, including the Mallampati classification, mouth opening, and thyromental distance.

### 2.1. Randomization

An independent anesthesiologist, who was involved only in randomization, performed this before anesthetic induction using a computer-generated random number table (Random Allocation Software ver. 1.0®, Isfahan University of Medical Sciences, Isfahan, Iran). The patients were allocated to a group that used a simple aluminum stylet (Covidien LLC, Mansfield, MA, USA; Group S, *n* = 142) or a lighted stylet (LIGHT WAY; Ace Medical, Goyang, Korea; Group L, *n* = 142) for tracheal intubation with a direct laryngoscope.

### 2.2. Anesthesia Induction

Patients were pre-medicated with intravenous 0.03 mg/kg midazolam. Standard monitoring procedures were carried out, including electrocardiograms and non-invasive blood pressure, pulse oximetry, and relaxometry. To induce general anesthesia, 1.2–1.5 mg/kg propofol was infused intravenously; remifentanil was administered intravenously using a target-controlled infusion by an Orchestra® infusion pump system (Fresenius vial, Brezins, France) with target concentrations of 2–4 ng/mL. Muscle paralysis was induced by injecting a muscle relaxant (0.6 mg/kg rocuronium). Positive-pressure mask ventilation was performed with the supplemental inhalation of 6–8% sevoflurane and continuous infusion of remifentanil. Direct laryngoscopy was performed using a Macintosh direct blade No. 3 (Welch Allyn, NY, USA) after the loss of all four twitches on the train-of-four stimuli. After intubation, anesthesia was maintained with a 2–3% sevoflurane and remifentanil infusion.

### 2.3. Experimental Procedure

The lights in the operating room were not dimmed. Intubation was performed with patients in the supine position and the head placed on a 7 cm-high hard pillow to achieve the conventional sniffing position. An endotracheal tube (TaperGuard™ cuff; Covidien, Mansfield, MA, USA) with an internal diameter of 7 mm was used for females and 7.5 mm tubes were used for males. Three anesthesiologists, who had each performed >1000 direct laryngoscopies, performed the direct laryngoscopy in this study. In both groups, the stylet-assisted tracheal intubation was performed with a direct laryngoscope without cricoid pressure, and the endotracheal tubes were bent 90° at the proximal part of the cuff balloon using a stylet [3]. A simple stylet was used in Group S. During the intubation, the endotracheal tube was inserted through the vocal cords with a stylet and the stylet was removed when the tip of the endotracheal tube was thought to pass the larynx. A lighted stylet was used in Group L and was removed after confirming transillumination of the soft tissue above the suprasternal notch (Figure 1). Successful intubation was defined as the correct placement of the endotracheal tube in the trachea, as confirmed by end-tidal CO_2_ capnometry, bilateral lung auscultation, and misting of the endotracheal tube. If the third attempt failed, that patient was considered a “fail” and an alternative technique was used for intubation by the duty anesthesiologist, such as video laryngoscopy or fiberoptic intubation.

The duration of each attempt was recorded as the time at which the laryngoscope was inserted to the time at which success or failure was confirmed. The total intubation time was defined as the sum of the duration of the total attempts (as many as three). Failed cases were not included in the determination of the total intubation time for either technique. The laryngoscopic view was documented using the Cormack-Lehane grade [4]: grade I (full view of the vocal cord), grade II (partial view of the vocal cord), grade III (only epiglottis visible), and grade IV (epiglottis not visible). Grade III–IV was defined as a difficult laryngoscopy. The success rate at the first attempt was compared between the two groups in a subgroup analysis.

After extubation, any blood in the oral cavity or on the endotracheal cuff was examined for evidence of mucosal bleeding by the attending anesthesiologist. The patients were then transferred to the post-anesthesia care unit. A postoperative sore throat was evaluated 1 and 24 h after surgery by a nurse blinded to the group assignments. The degree of postoperative sore throat was assessed using a numeric rating scale (0, no pain; 100, intractable pain). Other airway complications, such as dysphagia or dental injury, were queried and recorded.

### 2.4. Outcome Measures

The primary outcome was the success rate with the first intubation attempt. Secondary outcome measures were the total intubation time, the incidence of tinged blood at the tube tip after extubation, and the degree of sore throat at 1 hour and 1 day postoperative. A previous investigation reported that the success rate of laryngoscopic intubation with a simple stylet at the first attempt was 87% [5]. The increase in success rate at the first attempt to 98% by a lighted stylet was considered clinically significant. A sample size of 142 participants per group was calculated with a significance level of 0.05 (α = 0.05) and a power of 80% (β = 0.20) considering a 10% drop-out rate.

### 2.5. Statistical Analysis

SPSS 21.0 for Windows (SPSS Inc., Chicago, IL, USA) was used for the statistical analysis and normality testing. The Mann-Whitney U test was used for continuous variables (i.e., age, height, weight, mouth opening, thyromental distance, total intubation time, and the degree of postoperative sore throat) and the chi-square test or Fisher’s exact test was used for categorical variables (i.e., sex, ASA physical class, Mallampati class, success rate at the first attempt, total trial number, incidence of mucosal bleeding and other airway complications, and Cormack-Lehane grade). Data are presented as the median (interquartile range) or number (%). A *p*-value <0.05 was considered significant.

## 3. Results

We initially screened 298 patients for study enrollment. Of these, 12 patients refused to enroll in the study and two were excluded due to a change to spinal anesthesia. Ultimately, the study enrolled 284 patients, of whom 142 were assigned to Group S and the other 142 to Group L. The data from 284 patients were used for the final analysis (Figure 2). No significant differences were detected in patient characteristics, including age, weight, height, sex, ASA physical class, Mallampati class, mouth opening, or thyromental distance, between the two groups (Table 1).

The lighted stylet significantly increased the success rate of tracheal intubation at the first attempt compared to the simple stylet (128 (90%) vs. 140 (99%), Groups S and L, respectively, *p* = 0.003, Table 2). The total success rate was not different between the two groups (139 (98%) vs. 140 (99%), Groups S and L, respectively). Three patients in Group S (Cormack-Lehane grade III) and two patients in Group L (Cormack-Lehane grade III) were considered failures, and were intubated with a fiberoptic bronchoscope.

There were 19 (13%) difficult laryngoscopies in Group S and 23 (16%) in Group L. The success rate at the first attempt for these cases was significantly lower in Group S than in Group L (9 (44%) vs. 21 (90%), *p* < 0.001, Groups S and L, respectively).

The total intubation time did not differ significantly between the two groups. The incidence of mucosal bleeding was significantly higher in Group S than in Group L (35 (25%) vs. 19 (13%), *p* = 0.011, Groups S and L, respectively). No other adverse events (e.g., dysphagia, dental injury) were observed in either group. In addition, the degree of postoperative sore throat did not differ significantly between the two groups.

## 4. Discussion

Our data indicate that the lighted stylet is an effective and safe device for assisting intubation with a direct laryngoscope. The lighted stylet was associated with a higher success rate of first attempts at tracheal intubation and less mucosal bleeding.

The lighted stylet was initially used only as a bougie during oral or nasal intubation [6,7]. Biehl reported the first case of using a lighted wand during tracheal intubation with a direct laryngoscope [8]. The wand was threaded through the Murphy hole of an endotracheal tube, which was guided down over the tip of the stylet into the trachea. Wu et al. showed the usefulness of a direct laryngoscope during lighted wand intubation in patients with an unstable cervical spine [9]. In addition, Agro et al. demonstrated the effectiveness of a lighted wand during tracheal intubation with a direct laryngoscope in the manual inline stabilized position [10]. However, that study had no control group, so the superiority over other devices-including the simple stylet-was not investigated. The present investigation is the first randomized observational study conducted with a large number of patients undergoing general anesthesia to demonstrate the effectiveness of a lighted stylet compared to a simple stylet for tracheal intubation with a direct laryngoscope.

Esophageal intubation remains one of the most worrisome complications during blind intubation. Transillumination has been used to reliably distinguish between intratracheal and esophageal or intrapharyngeal intubation [6]. Transillumination of the suprasternal area means that the tip of the endotracheal tube has passed the vocal cords and is located in the trachea. In our study, the lighted stylet effectively revealed the position of the stylet tip in the trachea during direct laryngoscopy, and helped to avoid esophageal intubation or other tube displacements. Even in obese patients with a short neck (data not shown), transillumination of the neck area was sufficient to detect the tip of the tracheal tube in the ambient light. However, two failed cases occurred in Group L due to misjudged tube placement-the intubator believed faint illumination (esophageal or intrapharyngeal intubation) to be tracheal intubation. To avoid misjudgment, verification of the clear and bright transillumination on the trachea is indispensable and tube placement should be confirmed using capnography [11].

Tracheal intubation with a direct laryngoscope is a skill that needs considerable training, and alignment of the oropharyngeal-laryngeal axes is not always possible [12,13]. Although the use of a video laryngoscope increases the success rate of a difficult intubation, some investigators have reported low success rates at first attempts of 84–87% with the DCI videoscope under difficult simulated conditions [5,14]. In addition, success rates are lower in the emergency room (81%) [15] or intensive care unit (79%) [16] when a C-MAC is used, even for easier airways. In this study, the lighted stylet showed a high success rate (90%) for the first attempt, which was comparable with a video laryngoscopic device for difficult laryngoscopies. Therefore, we believe that rapid tracheal intubation can be achieved with a direct laryngoscope combined with a lighted stylet for a difficult laryngoscopy.

Failed intubation or repeated intubation attempts can cause various kinds of airway complications or critical problems. Rapidly securing the airway is much more important in emergency conditions. Orlando et al. reported that 11% of laryngoscopic intubations failed with one attempt [17], and the failure rate was much higher (26.1%) in the emergency department [15]. In our study, the failure rates at the first attempt were 10% in Group S and 1% in Group L. The higher failure rate with the simple stylet may be attributed to the obscured position of the tube tip-a shallow position of the tube tip may result in dislocation when removing the stylet. In contrast, the lighted stylet guides the operator to insert the endotracheal tube sufficiently deep into the trachea and avoid malpositioning of the tube. In addition, esophageal intubation is of great clinical significance. One study demonstrated that a single episode of esophageal intubation is associated with an increase in the incidence of hypoxemia, aspiration, cardiac dysrhythmias, and cardiac arrest [18]. Therefore, use of a lighted stylet may decrease the failure rate during the first laryngoscopic intubation in patients who need rapid airway control, such as during cardiopulmonary resuscitation or in patients with a poor lung reservoir (due to an inflammatory lung condition, pregnancy, or morbid obesity).

Fan et al. reported cases of airway injury after endotracheal intubation from misdirected endotracheal tubes [19]. In those cases, a rigid endotracheal tube with a stylet (or even a protruded stylet) caused severe airway trauma. Although laryngeal injury can be greatly reduced by improvements in training and equipment design, a misled endotracheal tube tip can still damage the larynx. The uncertainty of the tube tip position might result in traumatic movement or an excessively deep position of the endotracheal tube during stylet-assisted intubation, and these can lead to mucosal damage or severe tracheal injury. In contrast, our results show that the incidence of mucosal injury was lower in Group L than in Group S. We believe that the light-guided tube tip position enables a gentler movement of the endotracheal tube and helps to place the tube tip at a suitable depth.

Some limitations of our study should be mentioned. Firstly, the anesthesiologist responsible for intubation was not blinded to the assigned groups, which may have introduced bias. However, the patients and data analysts were blinded to the study groups. Secondly, this study included difficult- and easy-to-intubate patients. Further studies should re-evaluate the effectiveness of a lighted stylet only in difficult-to-intubate patients. Thirdly, mucosal damage occurred with one or two attempts, possibly due to the L-shaped endotracheal tube. A straight endotracheal tube can be inserted into the trachea without tracheal stimulation by the tube tip. However, the tip of the L-shaped tube may scratch the anterior wall of the trachea during tube insertion, even in easy cases, and this can cause unexpected mucosal bleeding. However, the mucosal bleeding was not severe in any case in this study; it was simply identified by the blood-tinged endotracheal tube. Fourthly, postoperative hoarseness was not checked for. Mencke et al. suggested that suboptimal intubation conditions such as cough, resistance to laryngoscope blade insertion, inadequate muscle paralysis, and limb movement are associated with a high incidence of postoperative hoarseness and laryngeal injuries [19]. Because the tracheal intubations were performed under optimal conditions in this study, the postoperative complications examined focused on the incidence of mucosal damage and the severity of sore throat. A fifth limitation was that the incidence of difficult laryngoscopy was higher than in other reports. According to the study protocol, we did not perform any additional maneuver to improve the laryngeal view, which explains why the incidence rates of difficult laryngoscopy and failure were high. However, the incidence rates of difficult laryngoscopy did not differ between the two groups. Finally, this study was performed in ambient light. Ambient light can cause less-definite transillumination compared to working with the lights off. However, we had no difficulty detecting the trans-illuminated soft tissue, as supported by the high success rate of first attempts. In addition, laryngoscopy is generally performed under ambient light, so our results may better reflect reality in daily practice.

## 5. Conclusions

In conclusion, the lighted stylet increased the initial success rate of tracheal intubation by direct laryngoscopy via transillumination of the trachea. This indicates that a lighted stylet can be an effective substitution for a simple stylet during laryngoscopic intubation.

## Figures and Tables

**Figure 1 jcm-08-00140-f001:**
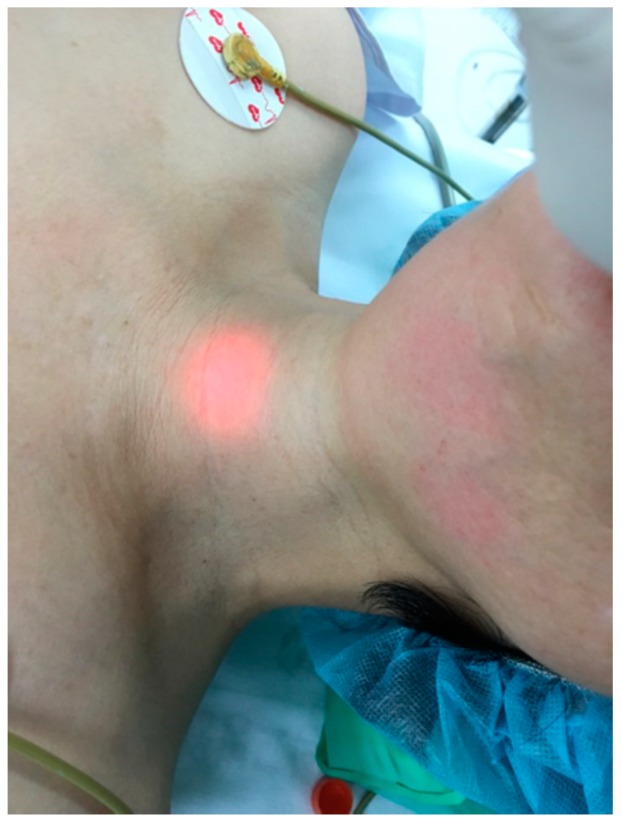
Transillumination of the suprasternal soft tissues by the lighted stylet.

**Figure 2 jcm-08-00140-f002:**
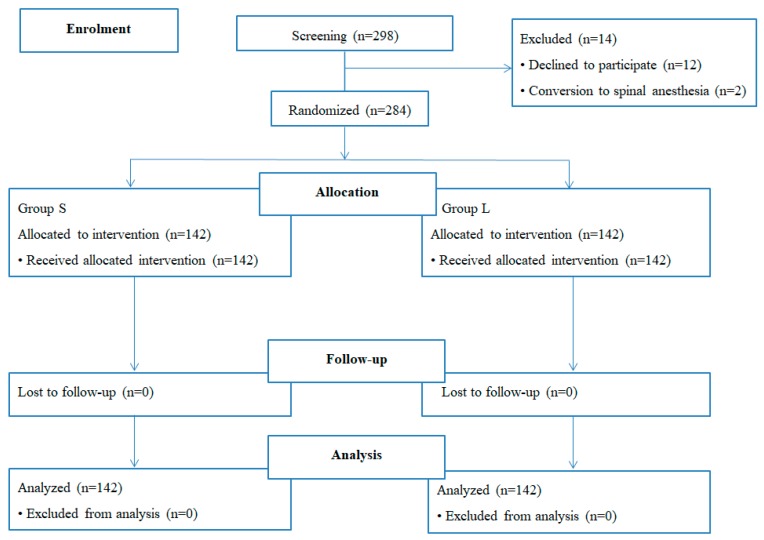
CONSORT diagram for the trial.

**Table 1 jcm-08-00140-t001:** Demographic data and preoperative airway evaluation.

	Group S (*n* = 142)	Group L (*n* = 142)	*p*-Value
Age (years)	52 (41–60)	51 (42–59)	0.642
Height (cm)	163.0 (156.5–169.6)	160.7 (155.8–170.0)	0.767
Weight (kg)	62.3 (54.4–71.6)	61.8 (54.4–74.1)	0.925
Gender (M/F), n (%)	61 (43)/81 (57)	57 (40)/85 (60)	0.718
ASA class (1/2), n (%)	73 (51)/69 (49)	84 (59)/58 (41)	0.233
Mallampati (I/II/III/IV), n (%)	72(51)/43(30)/25(18)/2(1)	74(52)/51(36)/15 (11)/2(1)	0.361
Mouth opening (mm)	50 (40–54)	50 (40–50)	0.687
Thyromental distance (mm)	90 (80–100)	90 (85–100)	0.543

Data are expressed as median (interquartile range) or numbers (%).

**Table 2 jcm-08-00140-t002:** Intubation parameters and perioperative complications.

		Group S (*n* = 142)		Group L (*n* = 142)	*p*-Value
Cormack-Lehane grade (I/II/III/IV)		79/44/19/0		83/36/23/0	0.527
Total success rate at 1st attempt, n (%)		128 (90)		140 (99)	0.003
Number of attempts (1/2/3/F)		128/9/2/3		140/0/0/2	0.002
Intubation time (s)		20.0 (17.0–24.5)		22.0 (19.0–25.0)	0.094
Mucosal bleeding			35 (25)	19 (13)	0.011
VAS (PACU)		15 (0–40)		20 (0–30)	0.957
VAS (POD#1)		10 (0–20)		0 (0–15)	0.338

Data are expressed as median (interquartile range) or numbers (%). VAS: visual analogue scale. PACU: postanesthesia care unit. POD#1: postoperative 1 day.

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
