# Peer review of "Comparison of Simple Stylet versus Lighted Stylet for Intubating the Trachea with a Direct Laryngoscope: A Randomized Clinical Trial"

_jcm, 2019, doi:10.3390/jcm8020140_

Reviewer 1 Report

It was a pleasure to read your manuscript. it is written in in a good style and should be easy to understand for the readers.

I have the only question:   

In anesthesia induction section 76 row:" Anesthesia was maintained with supplemental inhalation of 6‒8% sevoflurane" needs clarification Is this for anesthesia induction or for maintenance?

Scientific and clinical actuality - there is no scientific content in this manuscript at all. In current anaesthesiology clinical practice there are many options to anaesthetist for visualized tracheal intubation such as optical (airtraq), video (c-mac, glidescope) laryngoscopes and fiber-optic technique, moreover conventional laryngoscopes have a lot of modifications starting with light sources (halogen, led) and finishing with different types of blades.

Overall outcome in this study was similar between study groups, however lighted stylet group showed better profile in attempts needed for successful tracheal intubation and less side effects.

Methodology of this study raises some questions about inclusion criteria and especially about type of surgery chosen. At least does every patient eligible to participate in the study was randomized?

Manuscript is written in easy understandable style for readers and results are presented appropriately. However in result's comparisons between groups some group S results are presented first and group L results after but other results are presented group L first and group S after- it can mislead readers.

Author Response

I have the only question:  

In anesthesia induction section 76 row:" Anesthesia was maintained with supplemental inhalation of 6‒8% sevoflurane" needs clarification Is this for anesthesia induction or for maintenance?

 6-8 vol% sevoflurane was used only for the induction period. During the maintenance, 2-3 vol% sevoflurane was used. We modified the method part as bellows:

‘Positive-pressure mask ventilation was performed with supplemental inhalation of 6‒8% sevoflurane and continuous infusion of remifentanil. After intubation, anesthesia was maintained with 2-3% sevoflurane and remifentanil infusion.’

 Methodology of this study raises some questions about inclusion criteria and especially about type of surgery chosen. At least does every patient eligible to participate in the study was randomized?

Thank you for your comment. The patients were randomly divided into two groups using computer-generated random number table.

 Manuscript is written in easy understandable style for readers and results are presented appropriately. However in result's comparisons between groups some group S results are presented first and group L results after but other results are presented group L first and group S after- it can mislead readers.

Thank you for your comment. We revised the manuscript of abstract and result part as bellows:

‘The incidence of mucosal bleeding was significantly higher in Group S (35 (25%) vs. 19 (13%), P: 0.011, Group S and L, respectively).’

‘The success rate at first attempt for these cases was significantly lower in group S than in group L (9 [44%] vs. 21 [90%], P < 0.001, groups S and L, respectively).’

‘The incidence of mucosal bleeding was significantly higher in group S than in group L (35 [25%] vs. 19 [13%], P: 0.011, groups S and L, respectively).’

Reviewer 2 Report

This randomized controlled study compares two aids to tracheal intubation - plain stylet and lighted stylet. 

Major comments:

Some points have to be clarified or added -

The authors stated that all patients were checked for routine preoperative airway assessment. They mentioned Mallampati test, thyromental distance and mouth opening. Was neck extension also checked? Did the authors divide the patients to the groups according to expected difficulty of intubation? If so, why was for predicted difficult to intubate patients chosen classical direct laryngoscopy and not other more appropriate methods? (awake fiberoptic, video-laryngoscopy).

90% success rate with a plain stylet (it means 10% failure rate) on the first attempt seems quite high to me, the incidence of difficult intubation in the operating room in the literature is described between 1-3% (Paix, 2005). This should be explained in the discussion.

Were any additional maneuvers in order to improve the laryngeal view during laryngoscopy allowed in this trial? If so, this should be added to the manuscript - head re-positioning, external laryngeal manipulation (BURP), etc.

What was the consequence of "unsuccessful" attempt? How was that diagnosed? - etc. esophageal intubation, absence of capnography track...

The authors assessed postoperative adverse effects of tracheal intubation but some of them are missing. The incidence of postoperative hoarseness should be reported - it could be as high as 6.7%. Also zero incidence of dysphagia in this relatively large cohort seems to be rather surprising (see 6.4% incidence in another trial) (Rios, 2014).

The incidence of difficult laryngoscopy in this group is very high! The authors report overall almost 15% incidence of difficult laryngoscopy while the published incidence is up to 3% in mixed surgical population (except of obstetrics) - this should be explained.

Minor comments:

P2,L71 - add relaxometry to the monitoring of the patients

P2,L72 - replace "peripheral oxygen saturation tests" with "pulse oximetry"

P2,L78 - direct blade No3? - was it Macintosh design of the blade or other?

P3,L103 - the authors state that "difficult laryngoscopy was defined as CL score more than 3" - This should be clarified "3 or 4" is usually described as difficult laryngsocopy or even sometimes grade 2b as well.

P3,L111 - why temporary/long-lasting hoarseness was not studied as an adverse effect?

 Author Response

The authors stated that all patients were checked for routine preoperative airway assessment. They mentioned Mallampati test, thyromental distance and mouth opening. Was neck extension also checked? Did the authors divide the patients to the groups according to expected difficulty of intubation? If so, why was for predicted difficult to intubate patients chosen classical direct laryngoscopy and not other more appropriate methods? (awake fiberoptic, video-laryngoscopy).

 Neck extension was not included in the preoperative airway assessment. And the patients were randomly divided into the two groups, not according to expected difficulty of intubation. Fiberoptic bronchoscopy or videolaryngoscopy was prepared before induction and used only in the failed cases.

 90% success rate with a plain stylet (it means 10% failure rate) on the first attempt seems quite high to me, the incidence of difficult intubation in the operating room in the literature is described between 1-3% (Paix, 2005). This should be explained in the discussion.

 According to the study protocol, we did not perform any additional maneuver such as cricoid pressure to improve laryngeal view. This may result in increasing incidence of difficult laryngoscopy and failure rate.

We added in the limitation as bellows:

‘Fourth, the incidence of difficult laryngoscope is higher than other reports. According to the study protocol, we did not perform any additional maneuver to improve laryngeal view. This is why the incidence of difficult laryngoscopy and failure rates are high. However, the incidences of difficult laryngoscopy are not different between the two groups.’

 Were any additional maneuvers in order to improve the laryngeal view during laryngoscopy allowed in this trial? If so, this should be added to the manuscript - head re-positioning, external laryngeal manipulation (BURP), etc.

 We did not apply any additional maneuver according to the study protocol.

 What was the consequence of "unsuccessful" attempt? How was that diagnosed? - etc. esophageal intubation, absence of capnography track...

The consequence of unsuccessful attempt was intraoral intubation or intraesophageal intubation. That was diagnosed as ‘the absence of end-tidal CO2 capnometry, bilateral lung sound and misting of the endotracheal tube’.

 The authors assessed postoperative adverse effects of tracheal intubation but some of them are missing. The incidence of postoperative hoarseness should be reported - it could be as high as 6.7%. Also zero incidence of dysphagia in this relatively large cohort seems to be rather surprising (see 6.4% incidence in another trial) (Rios, 2014).

 Postoperative hoarseness was not checked. We examined postoperative complications focusing on the mucosal damage and sore throat. We defined dysphagia as ‘difficulty in swallowing water 24 hrs after operation. There was no patient with difficulty in swallowing.

 The incidence of difficult laryngoscopy in this group is very high! The authors report overall almost 15% incidence of difficult laryngoscopy while the published incidence is up to 3% in mixed surgical population (except of obstetrics) - this should be explained.

  We agree with you. According to the study protocol, we did not perform any additional maneuver to improve laryngeal view. This is why the incidence of difficult laryngoscopy is higher than other reports. However, the incidences of difficult laryngoscopy are not different between the two groups.

 Minor comments:

P2,L71 - add relaxometry to the monitoring of the patients

à Thank you for your comment. We modified the method part as bellows:

‘Standard monitoring procedures were carried out, including electrocardiograms and noninvasive blood pressure, pulse oximetry and relaxometry.’

P2,L72 - replace "peripheral oxygen saturation tests" with "pulse oximetry"

à Thank you for your comment. We modified the method part as bellows:

‘Standard monitoring procedures were carried out, including electrocardiograms and noninvasive blood pressure, pulse oximetry and relaxometry.’

 P2,L78 - direct blade No3? - was it Macintosh design of the blade or other?

à Thank you for your comment. We modified the method part as bellows:

‘Direct laryngoscopy was performed using a Macintosh direct blade No. 3 (Welch Allyn, NY, USA) after the loss of all four twitches on the train-of-four stimuli.’

 P3,L103 - the authors state that "difficult laryngoscopy was defined as CL score more than 3" - This should be clarified "3 or 4" is usually described as difficult laryngsocopy or even sometimes grade 2b as well.

à Thank you for your comment. We modified the method part as bellows:

‘A grade III or IV was defined as a difficult laryngoscopy.’

 P3,L111 - why temporary/long-lasting hoarseness was not studied as an adverse effect?

We are sorry but postoperative hoarseness was not checked. We examined postoperative complications focusing mucosal damage and sore throat.

Reviewer 3 Report

Overall, interesting paper, simple design, and straightforward presentation. The idea of combining lighted stylet and DL is novel. Definitely needs further study and research. Suggest publishing the success and failures presented across Cormack-Lehane grades in both the groups. Intuitively, one would expect the lighted stylet to be more useful for higher CL grades.

For the control group, ie direct laryngoscopy with simple stylet, a first pass failure rate of 10% in a anesthesia practice for elective surgeries (with difficult patients excluded) and where most patients had Cormack-Lehane grade I or II views seems to be excessive

Why was the simple stylet bent to 90 degrees ? That may be the optimal bent angle for the lightwand but not for direct laryngoscopy with simple stylet. This may cause more trauma than the usual "hockey-stick" configuration, and may have increased the first pass failure rate.

I am not sure we can conclude that the lighted stylet is superior to the simple stylet in out-of-OR situations or where rapid airway control is required, as mentioned in the discussion (lines 204 to 206) based on this study (performed in a controlled anesthesia setting, with strict exclusion criteria)

Author Response

Overall, interesting paper, simple design, and straightforward presentation. The idea of combining lighted stylet and DL is novel. Definitely needs further study and research. Suggest publishing the success and failures presented across Cormack-Lehane grades in both the groups. Intuitively, one would expect the lighted stylet to be more useful for higher CL grades.

For the control group, ie direct laryngoscopy with simple stylet, a first pass failure rate of 10% in a anesthesia practice for elective surgeries (with difficult patients excluded) and where most patients had Cormack-Lehane grade I or II views seems to be excessive

Thank you for your comment. First pass failure rate of 10% may be due to a high incidence of the difficult laryngoscopy. According to the study protocol, we did not perform any additional maneuver (ie. cricoid pressure, head repositioning or jaw-thrust) to improve laryngeal view. This may be why the incidence of difficult laryngoscopy is higher than other reports. There are some reports indicating higher failure rate than 10%.  Seroki showed the 13% failure rate using direct laryngoscope and 10~13% even when videolaryngoscopy is used. (Seroki 2010) And the failure rates in ICU or OR were reported as 19-21%. (Sakles 2012, Noppens 2012).

 Why was the simple stylet bent to 90 degrees ? That may be the optimal bent angle for the lightwand but not for direct laryngoscopy with simple stylet. This may cause more trauma than the usual "hockey-stick" configuration, and may have increased the first pass failure rate.

L-shape is generally-used form in the intubation with simple stylet as bellows:

We described that the tube was bent 90° but the real shape of the endotracheal tube was hockey-stick like the above figure. We will change the description, if necessary.

 I am not sure we can conclude that the lighted stylet is superior to the simple stylet in out-of-OR situations or where rapid airway control is required, as mentioned in the discussion (lines 204 to 206) based on this study (performed in a controlled anesthesia setting, with strict exclusion criteria)

We don’t think that lighted stylet is helpful in all patients in ICU or ER. Recently, intubation in ICU or ER is usually done under anesthetic induction and muscle relaxation, which is very similar condition to OR. The lighted stylet, therefore, will be superior to simple stylet in terms of increasing success rate at 1st attempt and decreasing mucosal bleeding, when sedative and muscle relaxant were used.

 Round  2

Reviewer 2 Report

The authors responded sufficiently on my questions and comments. 

However, the manuscript should be checked cartefully for misspells and language errors. The grammar shoudl be corrected prior to publication.

In concordance with my previous report. I feel that the authors should report that they did not check hoarseness among the postoperative complaints and comment it in the discussion section of the manuscript. . This may be quite important mainly if rigid stylet is used during tracheal intubation. 

Author Response

The authors responded sufficiently on my questions and comments. 

However, the manuscript should be checked carefully for misspells and language errors. The grammar should be corrected prior to publication.

 Thank you for your comment. We have checked misspells and language errors via scientific editing service.

 We added the certificate from the editing service in the acknowledgement as bellows:

‘The English in this document has been checked by at least two professional editors, both

native speakers of English. For a certificate, please see:

www.textcheck.com/certificate/eQe8Gf’

 In concordance with my previous report. I feel that the authors should report that they did not check hoarseness among the postoperative complaints and comment it in the discussion section of the manuscript. . This may be quite important mainly if rigid stylet is used during tracheal intubation. 

 Thank you for your comment. We have added a limitation as bellows: 

‘Fourth, postoperative hoarseness was not checked for. Mencke et al. suggested that suboptimal intubation conditions, such as cough, resistance to laryngoscope blade insertion, inadequate muscle paralysis, and limb movement, are associated with a high incidence of postoperative hoarseness and laryngeal injuries [19]. Because the tracheal intubations were performed under optimal conditions in this study, the postoperative complications examined focused on the incidence of mucosal damage and severity of sore throat.’